# A Transparent, and Self-Healable Strain-Sensor E-Skin Based on Polyurethane Membrane with Silver Nanowires

Rundong Wang [1], Shuangjiang Feng [1], Yanyun Wang [1], Chengqian Li [1], Xiaohai Bu [2], Yuzhong Huang [3], Man He [1,*] and Yuming Zhou [1,*]

[1]  Jiangsu Optoelectronic Functional Materials and Engineering Laboratory, School of Chemistry and Chemical Engineering, Southeast University, Nanjing 211189, China; rdwang@seu.edu.cn (R.W.); shuangjiangchem@163.com (S.F.); 230198854@seu.edu.cn (Y.W.); lichengqian98@seu.edu.cn (C.L.)

[2]  School of Materials Science and Engineering, Nanjing Institute of Technology, Nanjing 211167, China; xhbu@njit.edu.cn

[3]  ZYfire Hose Co., Ltd., Taizhou 225599, China; china@zyfire.com

*  Correspondence: heman@seu.edu.cn (M.H.); ymzhou@seu.edu.cn (Y.Z.)

**Abstract:** Electronic skin (E-skin) is increasingly utilized in modern society, yet current E-skin technology suffers from issues, such as opacity, hardness, and fragility. To address these challenges, a novel E-skin was developed using polyurethane (PU) as the matrix material and silver nanowires (AgNWs) as the sensing material. By leveraging the small degree of microphase separation and lack of crystallization in the PU, combined with the appropriate length–diameter ratio of the AgNWs, the resulting E-skin exhibited a visible light transmittance of 75%. The E-skin also showed excellent self-healing properties (83.63% efficiency in the third repair) and mechanical properties (with almost no degradation after 60 tensile cycles) due to the reversible dynamic cross-linking network within the PU. The synergistic effect of PU and AgNWs resulted in exceptional sensing performance for the E-skin, with a gauge factor of 46 (when $\varepsilon$ = 10%). Moreover, the E-skin demonstrated signal stability during human joint motion monitoring and successfully identified different movement states, highlighting its potential for diverse applications. This research presents a simple yet effective approach for producing transparent, durable, and stable E-skin.

**Keywords:** electronic-skin (E-skin); strain-sensor; self-healing; polyurethane; AgNWs; spraying; transparent; UV blocking

## 1. Introduction

As societal development progresses, suboptimal health problems have become increasingly prominent in the population, and monitoring suboptimal health has garnered growing attention [1,2]. Electronic skin (E-skin), a novel wearable sensor, has found increasing application in areas such as motion tracking and health condition monitoring [3–5]. However, conventional sensors or E-skin suffer from drawbacks including unattractiveness, susceptibility to damage, and poor wearability, which impede the broad application of E-skin [6–8].

The aforementioned challenges have attracted the attention of researchers, who have made significant efforts towards their resolution. For instance, with regard to enhancing the transparency of E-skins, some researchers have drawn inspiration from ion conduction and developed a range of hydrogel-based E-skins that exhibit high transparency (with over 80% visible light transmittance) and good compatibility with human skin. Such skins have achieved a sensing gauge factor (GF) of at least ten within specific strain ranges [9,10]. However, hydrogels have a tendency to dry out, leading to durability issues, as well as reduced signal sensitivity at low deformation levels [11,12]. Other researchers have sought to enhance transparency by incorporating conductive nanomaterials into resins, achieving similar transparency levels in certain visible light bands [13,14]. Additionally,

some researchers have constructed conductive networks on the surfaces of E-skin substrates using materials such as silver nanowires (AgNWs), resulting in high transparency and sensitivity [15–17]. However, these approaches suffer from limitations, such as substrate hardness, poor sensing accuracy, partial transparency, and coloration. To overcome the challenge of low durability, researchers have explored the use of resin as a substrate for E-skins, which can improve their mechanical strength and longevity [18–20]. For instance, some researchers have used polyurethane (PU) as a substrate, on which conductive polymers or carbon nanotubes are deposited [21–24], or in situ polymerization of PPy-PDMS-PPy on a PU film [25]. Others have employed Polyvinylidene fluoride as a substrate, which is filled with conductive materials [26–29]. Furthermore, scholars have carried out extensive investigations on the determinants that affect the sensing efficacy of electronic skin, uncovering the impact of the concentration of conductive dopants and the ratio of length to diameter on sensing performance. This discovery furnishes a theoretical framework for addressing the predicament of electronic skin [30–32]. However, while these materials offer improved durability, they can be challenging to manufacture and may not satisfy aesthetic, comfort, and high-sensitivity requirements. Currently, the field of E-skin faces several outstanding issues that require resolution. These include the low sensitivity coefficient of stretch-sensing electronic skin at small strains [33–35], inadequate aesthetics [36–38], poor durability of the matrix material [37,39], stringent requirements for self-repair conditions [40], and suboptimal self-repair efficiency [41].

Although E-skin technology has made significant progress in various properties, there is a lack of reporting on E-skin materials that exhibit all of the following properties: transparency, durability, and self-healing. Therefore, the goal of this paper is to synthesize a multifunctional E-skin material that possesses transparency, durability, self-healing capabilities, and excellent sensing performance, and which can be widely utilized. In this study, we have successfully synthesized a novel E-skin with exceptional properties, including transparency, self-repair capability, and resistance to aging. This E-skin demonstrates a visible light transmittance exceeding 75%, a tensile strength of 30.67 MPa, an elongation at break of 302.79%, and a fracture toughness of 32.48 MJ·m$^{-3}$. After three cycles of fracture and repair, the electronic skin still exhibits a tensile strength of 25.62 MPa with a repair rate of 83.63%. Moreover, we have demonstrated that the mechanical properties of the E-skin remain practically unchanged after being subjected to 60 cycles of repeated stretching under different strains. Additionally, our results indicate that the synthesized E-skin exhibits superior sensing performance compared to other E-skins of its kind [17,42,43], with a gauge factor (GF) of 46 when $\varepsilon$ = 10%. The E-skin's stability during different human joint movement monitoring and its ability to recognize different motion states suggests excellent sensing performance. Previous research on electronic skin has tended to address specific issues, rather than comprehensively studying a multifunctional electronic skin possessing exceptional self-repair, durability, sensing, and transparency capabilities. To date, there are no reports in the literature of such a multifaceted electronic skin [21,43–46]. Furthermore, the remarkable ultraviolet shielding effect of such a comprehensive electronic skin suggests its enormous potential for real-time monitoring of human health and physical activity.

## 2. Materials and Methods

The current study describes a preparation method for an E-skin, which is presented schematically in Figure 1. Initially, a prepolymer is synthesized by reacting hexamethylene diisocyanate and polytetramethylene ether glycol. Subsequently, a dynamic reversible phenol–aminoformate bond is introduced into the PU matrix to fabricate the PU substrate material. In parallel, AgNWs with suitable length and diameter are prepared using a modified polyol method. Finally, the AgNWs dispersion is sprayed onto the PU surface via a spray coating method to generate the E-skin. By controlling the degree of microphase separation, the crystallinity of the PU, and the length and concentration of AgNWs, a transparent PU matrix with superior self-repair performance was achieved in the E-skin.

Additionally, the synergistic effect of PU and AgNWs, facilitated by the agglomeration and stacking of AgNWs, contributed to the E-skin's excellent sensing performance [47].

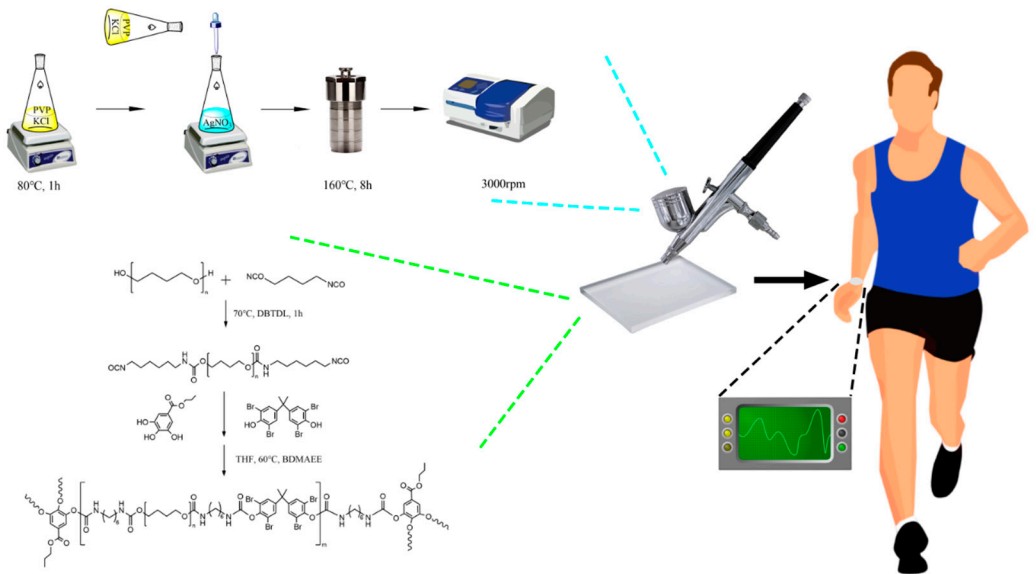

**Figure 1.** The process to produce the robust, transparent, and self-healable strain-sensor E-skin.

## 2.1. Materials and Chemicals

Hexamethylene diisocyanate (HDI, 99%), polytetramethylene ether glycol (PTMEG, Mn = 2000 g mol$^{-1}$), and dibutyltin dilaurate (DBTDL. 95%), were purchased from Aladdin (Shanghai) Reagent Co., Ltd., Shanghai, China. 3,5,3′,5′-Tetrabromobisphenol A (TBBPA, 98%), propyl gallate (PG. 99.87%), and bis (2-dimethylaminoethyl) ether (BDMAEE, 98%), were purchased from Bidepharm Chemical Reagent Co., Ltd., Shanghai, China. Tetrahydrofuran (THF, 99.5%), and polyvinylpyrrolidone (PVP, K90), were purchased from Macklin Reagent (Shanghai, China). Potassium chloride (KCl, 99.5%), ethylene glycol (EG, 99%), and silver nitrate (AgNO$_3$, 99.8%), were purchased from Sinopharm Chemical Reagent Co., Ltd., Shanghai, China. Anhydrous ethanol THF and EG were dehydrated with 4A molecular sieve for at least 24 h before use. PTMEG was vacuum dried at 120 °C for 2 h before use.

## 2.2. Preparation of PU

The synthesis of PU is divided into two stages. The prepolymer was synthesized by the reaction of PTMEG 2000 (32.00 g) and HDI (8.28 g) under the catalysis of DBTDL (1.2 mg) at 65 °C for 1 h with the protection of pure N$_2$. Then, we waited for the prepolymer to cool down to 60 °C, and dissolved TBBPA (12.30 g), PG (2.34 g), and BDMAEE (0.04 g) in 25 mL of anhydrous THF. The above solution is added to the prepolymer and stirred for 20 min at 60 °C. The polymer was cast onto a glass plate to form a film in an air-drying oven at 60 °C for 8 h. The film was further dried under vacuum oven at 60 °C for 72 h. Finally, the film, to meet the requirements was prepared, which has the following advantages: colorless, transparent, UV protection, high toughness and thermal repair. The film was kept in a desiccator for further use.

## 2.3. Preparation of AgNWs

AgNWs were synthesized using a modified polyol method. PVP (0.588 g) and KCl (0.514 g) were added to a conical flask with EG (20 mL) and completely dissolved by magnetic stirring under 80 °C for 1 h. AgNO$_3$ (0.34 g) was added to a beaker with EG (30 mL), and completely dissolved by magnetic stirring in the dark. After the AgNO$_3$ was dissolved, the PVP solution was added to the solution. After being stirred for 1 min, the mixture was transferred to a reaction kettle and heated at 160 °C for 7 h. After being cooled



to room temperature, the suspension was centrifuged 3 times for 15 min at 3000 rpm with ethanol [48,49]. Finally, AgNWs with a diameter of approximately 80 nm and a length of approximately 15~20 μm were dispersed in ethanol for further use.

### 2.4. Preparation of the Strain-Sensor E-Skin

The prepared ethanol dispersion of AgNWs was diluted to about 0.5 g·L$^{-1}$, and then was added to the spray gun. PU film was placed 30 cm under the infrared baking lamp. PU film surface was coated with AgNWs using a spray gun, the spraying rate is 5 cm·s$^{-1}$, each spraying interval is 10 s, and the spraying volume is about 0.3 mg·cm$^{-2}$. The transparent and self-healing strain-sensor E-skin (TSS) was obtained after resting for 2 h.

### 2.5. Measurements and Characterizations

The surface topography of PU films, AgNWs, and TSS film were performed using scanning electron microscopy (SEM, Inspect F50, FEI Company, Hillsboro, OR, USA), and transmission electron microscope (TEM, tecnai F20, FEI Company, Hillsboro, OR, USA) with element mapping analysis (EDS, X-MAX 80T, Oxford Instrument, London, UK). Optical pictures were obtained using a smart phone (Redmi K30, Xiaomi Inc., Beijing, China). The vibration of functional group was disclosed using the Attenuated Total Reflectance-Fourier transform Infrared spectroscopy (ATR-FTIR, Alpha II, Bruker Corporation, Karlsruhe, Germany) with an ATR attachment. X-ray photoelectron spectroscopy (XPS) analysis was carried out using a X electron spectrometer (Nexsa, Thermo Fisher Scientific Inc., Waltham, MA, USA). Check the chemical nature at a 100 W AlKα X-ray source and a base pressure of about $4 \times 10^{-9}$ mbar. The optical properties of PU film and TSS film were recorded using a UV-Vis-NIR spectrophotometer (UV-3600, Shimadzu Corporation, Kyoto, Japan). The thermal stability of samples was exhibited u thermogravimetric analysis (TGA, TG209 F3, Netzsch Gerätebau GmbH, Selb, Germany). The crystalline form of PU, and AgNWs was identified using X-ray diffraction (XRD, Ultima IV, Rigaku, Tokyo, Japan). The mechanical properties of films were tested via the uniaxial stretching machine (KJ-1065B, Guangdong Kejian Instrument Co., Ltd., Dongguan, China). The water contact angle of coatings was measured using a contact angle goniometer (OSA 100, LAUDA Scientific GmbH, Germany) with a 5 μL DI water droplet at room temperature. The resistance varied with tensile of the wearable sensor was gauged using Digital Multimeter (ET4401, Hangzhou East Tester Electronics Co., Ltd., Hangzhou, China). The initial resistance of the sensor and the resistance in case of strain were recorded as $R_0$ and R, respectively, and the normalized resistance was defined as $(R - R_0)/R_0$ for further explanation.

## 3. Results and Discussion

### 3.1. Measurements and Characterizations of Flexible Film

To determine the structure of the films, SEM and TEM images of PU films, AgNWs and strain-sensor E-skin are shown in Figure 2. The length of the synthesized AgNWs was about 15~20 μm and the diameter was about 80 nm. The SEM images (Figure 2c,d) show that the AgNWs were well loaded on the surface of the PU film, and the AgNWs were uniformly distributed with a conductive layer thickness of about 0.3 μm. The diffraction images of AgNWs (Figure 2f) show that the lattice of AgNWs is obvious, and the TEM-EDS elemental mapping of AgNWs (Figure 2g) demonstrates the uniform distribution of silver elements. The above two points prove the smooth growth of AgNW crystals [50].

In the ATR-FTIR spectrum of the film (Figure 3a), the absorption peak attributed to the isocyanate groups (about 2274 cm$^{-1}$ in the prepolymer) disappeared completely, indicating a complete reaction of the isocyanate groups. While the stretching vibration of the non-hydrogen bonded C=O of the urethane groups in the prepolymer at 1720 cm$^{-1}$, blue shifted to 1752 cm$^{-1}$ for the TBBPA chain extender, indicating that this shift is due to the electrical attraction effect of adjacent bromine groups. The peak at 742 cm$^{-1}$ was a characteristic band for the TBBPA chain extender. The peak at 1364 cm$^{-1}$ was a characteristic band for the PG cross-linker. The weak absorption band at 3331 cm$^{-1}$ was caused by N-H stretching

vibrations. The weak peaks at 2938 cm$^{-1}$ and 2859 cm$^{-1}$ were attributed to symmetric and asymmetric vibrations of methyl and methylene groups. The band at 1454 cm$^{-1}$ was considered to be a vibration of the benzene ring [51,52].

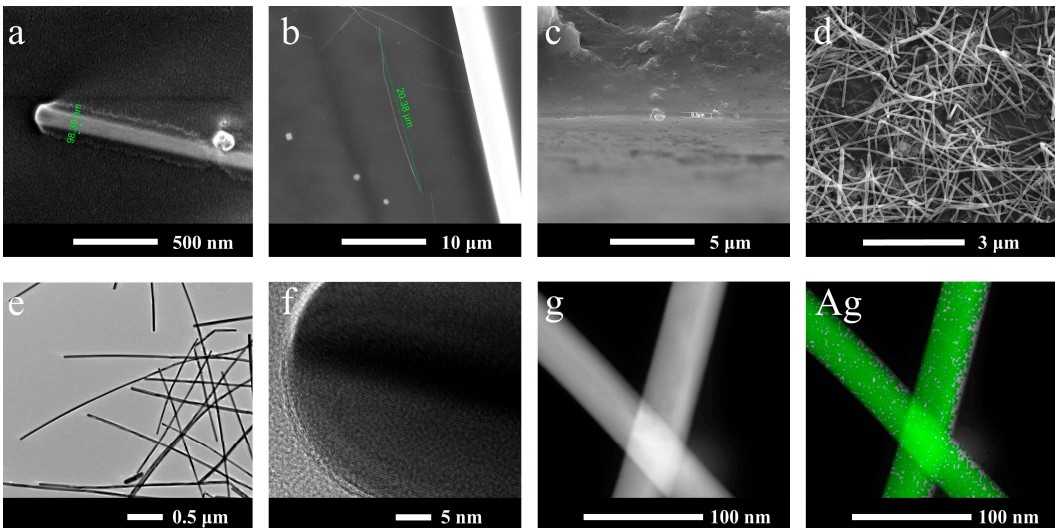

**Figure 2.** (**a**,**b**) SEM images of AgNWs; (**c**) Lateral morphology of AgNWs loaded on PU film; (**d**) Overall morphology of TSS; (**e**) TEM image of AgNWs; (**f**) Diffraction images of AgNWs; (**g**) TEM-EDS elemental mapping of AgNWs.

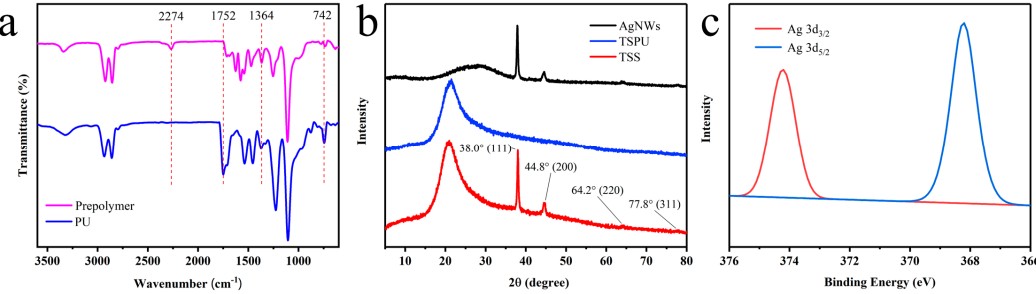

**Figure 3.** (**a**) FTIR spectra of PU, prepolymer; (**b**) XRD of PU, AgNWs and TSS; (**c**) XPS of TSS.

A comparison of the three curves reveals the characteristic diffraction peaks of the AgNWs. The peak observed at 2θ = 21.7° is attributed to PU, while the peaks at 38.0°, 44.8°, 64.2° and 77.8° correspond to scattering from the (111), (200), (220) and (311) crystal planes, respectively, as confirmed by comparison with the standard card JCPDS No. 87-0719 face-centered cubic Ag. Notably, no other heterogeneous peaks were observed, indicating high purity of the AgNWs. The narrow and high peak at 38.0° suggests that the AgNWs primarily grow towards the (111) crystal plane, indicative of a great length–diameter ratio, consistent with SEM and TEM results. The attachment of AgNWs to the PU surface is attributed to a physical bond, as evident from the lack of peak position shifts in the XRD test results [17]. Figure 3c shows the X-ray photoelectron spectroscopy (XPS) of TSS. The binding energy peaks at 368.2 eV and 374.2 eV are assigned to Ag 3d$_{5/2}$ and Ag 3d$_{3/2}$, respectively, indicating that the AgNWs are unoxidized silver monomers. This suggests that the AgNWs are in a metallic state rather than an oxidized state [53,54].

As shown in Figure 4a, the polyurethane film is colorless and transparent with a light transmission greater than 90% at 400~800 nm and with a film thickness of about 0.5 mm. Additionally, the PU films have excellent UV resistance (300~400 nm) due to the presence of benzene rings, as shown in Figure 4a. TSS film was also transparent with a light transmission greater than 75% at 400~800 nm and maintains the UV resistance. Optical

photos (Figure 4c,d) show the transparency of PU and E-skin under practical application conditions. The patterns can be clearly seen through PU and E-skin, which can meet the aesthetic needs in practical applications.

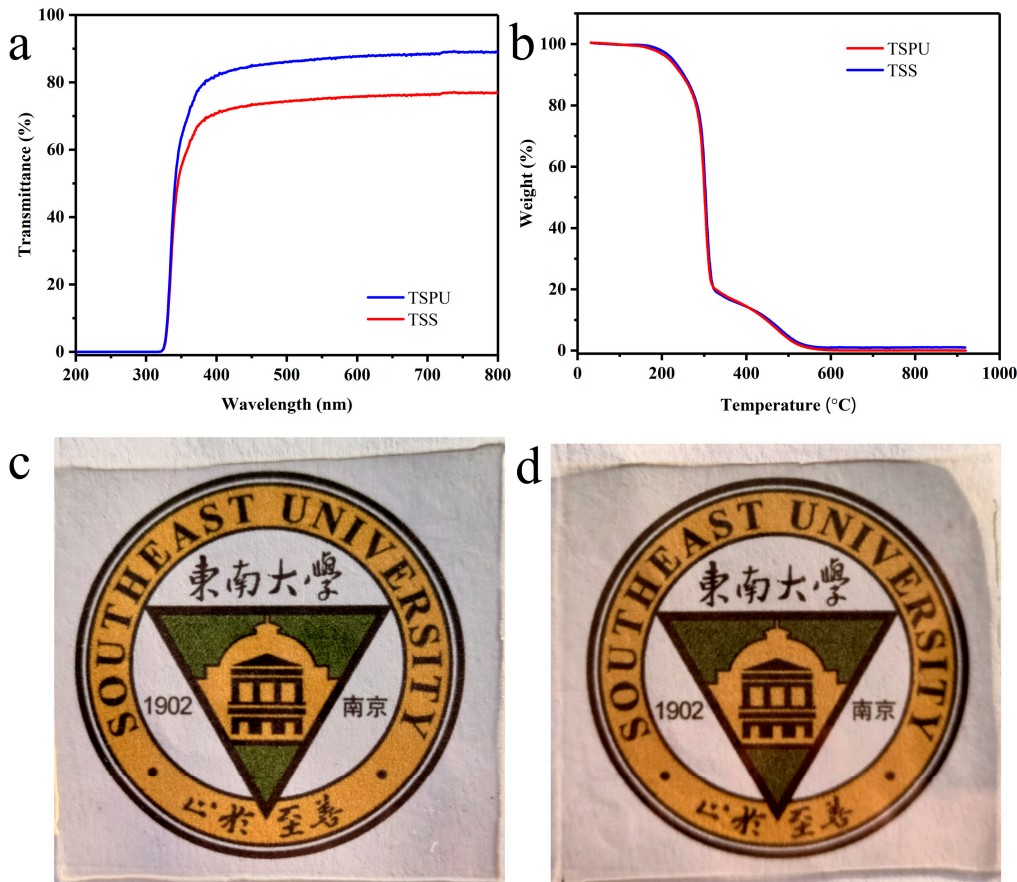

**Figure 4.** (**a**) UV–Vis transmission spectra of PU, TSS films in the range of 200–800 nm; (**b**) TGA of PU, TSS; (**c**) The optical image of PU; (**d**) The optical image of TSS.

Thermogravimetric analysis was used to determine the thermal stability of the flexible conductive film (as shown in Figure 4b). The TG curves of PU film show three main stages of degradation. The first stage (<170 °C) is due to the volatilization of low molecular substances, such as water and THF. The hydrogen bonds within the PU are opened and the hard chain segments are decomposed, which forms the second stage (170~325 °C). The third stage (>325 °C) is related to the decomposition of the soft chain segments. Since the main component of the soft laminated film was PU, the appearance of the decomposition curve was the same except for one aspect, the residual weight after the decomposition of TSS was 1.02%, which is caused by the incomplete decomposition of the oxidation of AgNWs. Considering that as a wearable device the sensor generally works at room temperature, the stability of the conductive soft film is ensured.

The water contact angle of PU is shown in Figure 5a, where it was found to reach 105.22° due to the presence of hydrophobic groups in PU. The water contact angle of TSS was also tested at different times and is shown in Figure 5c. While the contact angle decreased with time, it remained above 106°, indicating incomplete wetting behavior due to the nanoscale nature of the AgNWs, resulting in a hydrophobic surface on the sensor. To further evaluate the fouling resistance of the E-skin, we applied drops of milk, coffee, cola, and juice onto the E-skin and recorded optical images after 10 s (Figure 5b). The droplets retained their shape, demonstrating that they were easily removed from the surface of the E-skin.

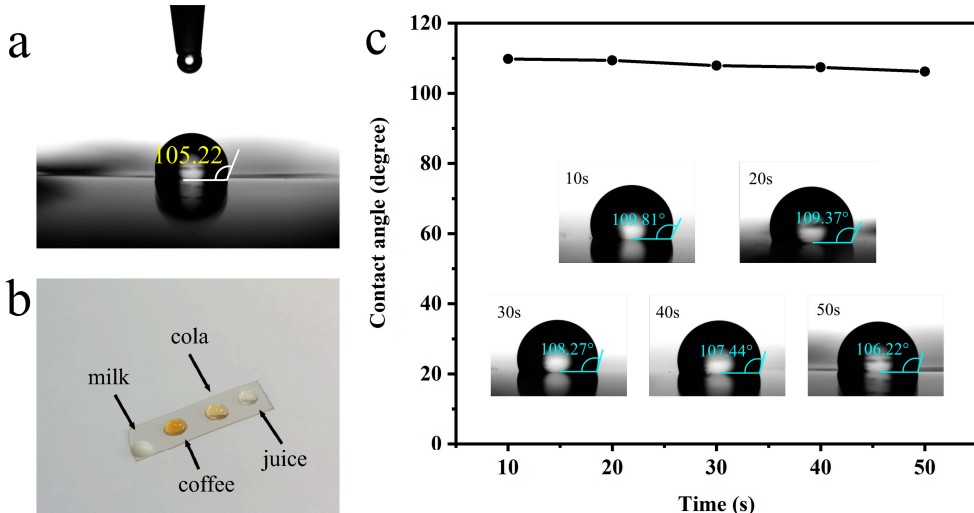

**Figure 5.** (**a**) Water contact angle of PU; (**b**) The shape of milk, coffee, cola and juice on the E-skin after dropping for 10 s; (**c**) Variation of water contact angle with TSS film time on.

The mechanical properties of the films were measured using uniaxial stretching tests (specifications: length 12.5 mm, width 2 mm, and thickness 0.5 mm) at ambient conditions (25 ± 2 °C, 60 ± 10% RH) at a stretching rate of 50 mm/min. The stress–strain curves of the PU films are shown in Figure 4b. The PU films exhibited good self-healing ability due to the reversibility of the phenol carbamate bonds of TBBPA and PG at high temperatures. In the optical microscope images, the cross scratches of PU almost disappeared after healing at 100 °C for 2 h (Figure 6a). The PU samples could be recycled at least three times after only 10 min of hot pressing at 100 °C (10 MPa) (Figure 6b). The tensile strength decreased slightly due to the irreversible reaction of the deblocked isocyanate with water and the elongation at break increased accordingly. Even so, the recovered sample after three times remained 83% of the original sample (Figure 6b).

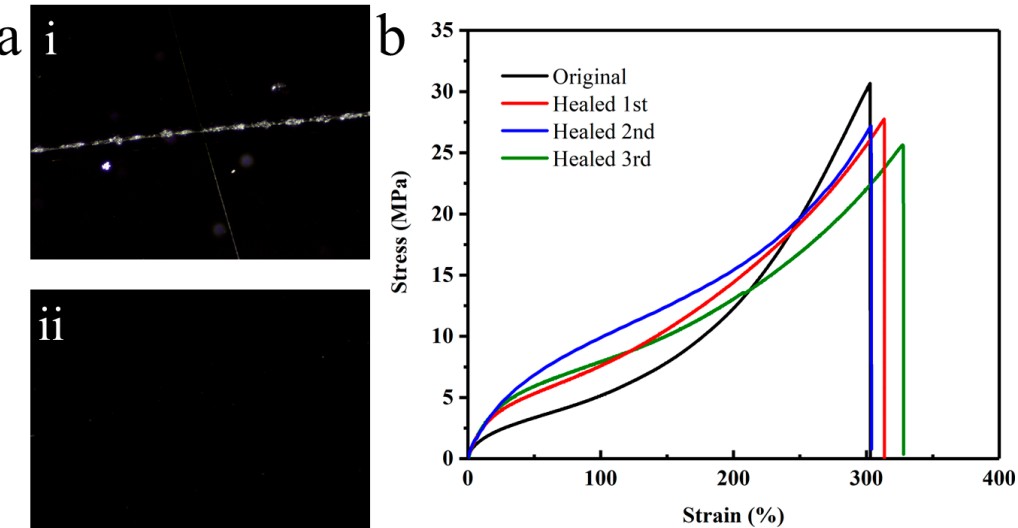

**Figure 6.** (**a**) Optical microscope images of TSS which were (**i**) scratched and (**ii**) healed; (**b**) Stress–strain curves of recycled TSS for three times.

The cyclic loading unloading tests with 10% (Figure 7a) strain and 100% (Figure 7b) are shown in Figure 7. The results show the mechanical curves of TSS films at 2 constant elongations with 60 repetitions of stretching. When the film recovers, its mechanical curve lags behind the mechanical curve during stretching. The reason is that the film

was subjected to elastic deformation (including fast elastic deformation and slow elastic deformation) and plastic deformation caused by tensile deformation, and the film did not return to the initial state in time when it recovered. It is also evident that the mechanical curve of the first stretching was significantly different from that of the subsequent stretching. The slightly lower recovery in the first cycle was due to the breakage of the hydrogen bond in the first stretch, while the crosslink established by the phenol–carbamate bond remained unchanged in the subsequent cycles. It could be seen that after the first stretch, there was some degree of film relaxation, and the subsequent stretch/recovery curves were close, indicating that subsequent stretches have little effect on film relaxation.

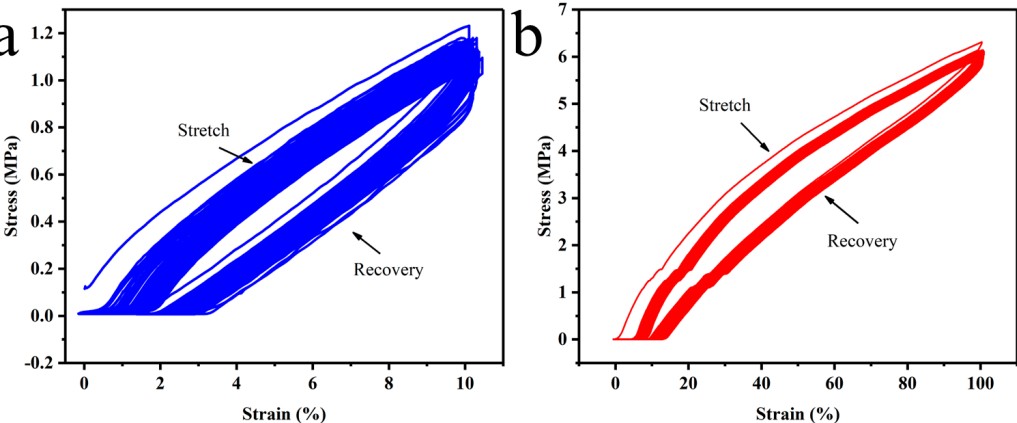

**Figure 7.** Strain-load curves for AgNWs with constant strain (**a**) 10% and (**b**) 100% repeated stretching.

### 3.2. Performance Testing of the E-Skin

The sensing performance of the E-skin when stretched is displayed in Figure 8. The gauge factor (GF) was defined as $(\Delta R/R_0)/\varepsilon$ (where $R_0$, $\Delta R$, and $\varepsilon$ represent the initial resistance, change in resistance, and elongation, respectively.) The GF represents the sensitivity of the strain-sensor E-skin and is the most important performance indicator. Decreasing the initial resistance ($R_0$) and increasing the change in resistance ($\Delta R$) are two useful ideas to promote the sensitivity of the sensor. The normalized resistance ($\Delta R/R_0$) varies with elongation (Figure 8a). At initial stretching, electrical conduction was largely dependent on the AgNWs. Having many contact points ensured that there are sufficient conductive paths to reduce the initial resistance. As the rate of stretching increases, the resistance of the sensor increased rapidly. Since the continuity of the conductive network decreased significantly with increasing elongation, the normalized resistance of the sensor increases rapidly, and the GF increases as well. As calculated in Figure 8a, the GF of our prepared strain-sensor E-skin was 46 (when $\varepsilon$ = 10%). Compared with many related studies, the GF of the strain-sensor E-skin without complex structure reported so far is mostly less than 30 at the same strain. Therefore, the GF of the strain-sensor E-skin prepared using this method is better than many similar sensors. The fatigue resistance represents the reproducibility of the normalized resistance to the same strain under multiple stretch/release cycles. Because wearable strain sensors need to accommodate complex dynamic strains, it is an important factor to consider when designing sensors. When $\varepsilon$ = 10%, the test cycle is 250 times, as shown in Figure 8c. We found that the normalized resistance of the sensor is reproducible under cyclic tensile strain. Through statistical analysis of the normalized resistance after 250 cycles, it was observed that all values were between 3.85 and 4.0 at the maximum value. At the maximum value, 90% of the values were in the range of 0~0.1, while approximately 10% of the values deviated towards 0.15 due to limitations of the universal material testing machine in accurately changing direction at extreme values. Further analysis was conducted on six cycles, as shown in Figure 8d, revealing that the normalized resistance of the electronic skin oscillated between 0 and 4, with a mean of 0.05 for the lowest value and a mean of 3.95 for the highest value, indicating

good stability. In addition, a good strain-sensor E-skin should have good signal stability under different strains to face the complex working environment. Figure 8b showed the step-by-step test at different elongation rates, respectively. The normalized resistance curve of the sensor shows a plateau at a certain strain, which indicates that the signal stability of the strain-sensor E-skin is reliable (Figure 8b).

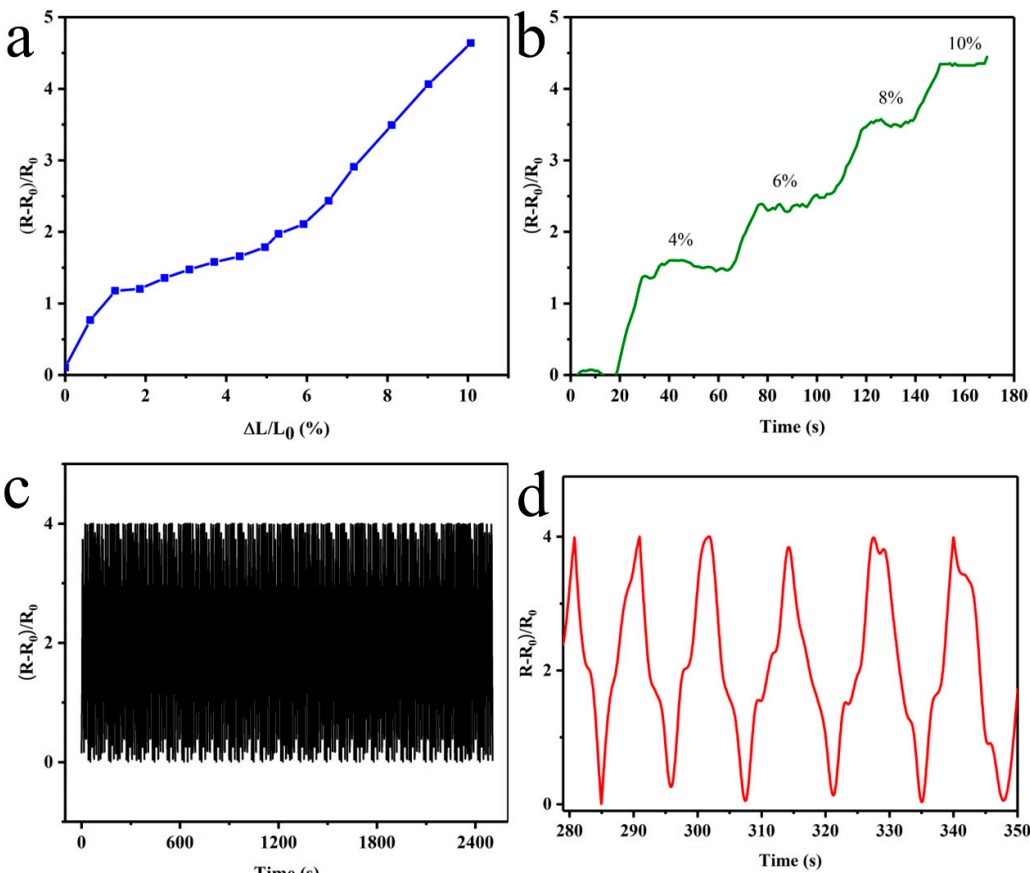

**Figure 8.** Strain sensing properties test (**a**) sensitivity; (**b**) step-by-step test; (**c**,**d**) fatigue resistance.

Strain-sensing E-skins were used for motion tracking and health monitoring due to their excellent sensitivity, fatigue resistance and signal stability. In the experiments, strain-sensing E-skins were placed on the finger (Figure 9a), wrist (Figure 9b) and knee (Figure 9c) to determine application performance. The monitoring of joint motion was exemplified by the finger application, where the standardized resistance of the E-skin was minimal at the beginning. As the finger was gradually bent to 90°, the E-skin was simultaneously bent and stretched, the conductive network was affected, and the normalized resistance increased to a maximum. As the finger was stretched, the normalized resistance gradually returned to its original value. Over multiple cycles, the normalized resistance was only slightly different due to the subtle differences in finger motion, showing good repeatability. The application on the wrist and knee was similar to that on the finger. It is noteworthy that we simulated walking with small and slow movements, and they showed significant differences in the curves compared to running with large and fast movements.

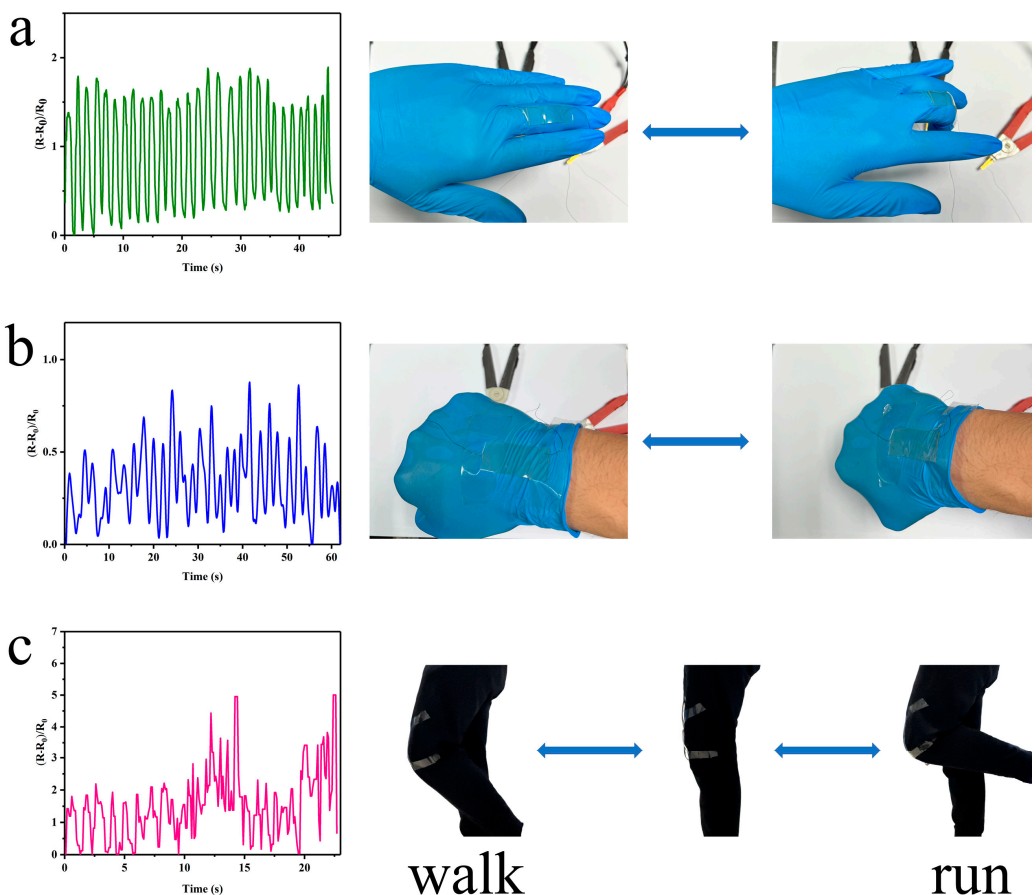

**Figure 9.** Application properties test put on (**a**) finger; (**b**) wrist; and (**c**) knees.

## 4. Conclusions

In conclusion, we have successfully synthesized a highly transparent and UV-resistant PU film, along with AgNWs of an optimal length–diameter ratio. These components were integrated using a simple method to create a strain-sensor E-skin with a conductive network on the surface of the PU film. The E-skin fabricated in this study exhibits excellent optical properties, self-healing abilities, and fatigue resistance owing to the molecular-level design of the substrate material. The appropriate length–diameter ratio of the synthesized AgNWs leads to extensive contact area between the nanowires coated on the PU surface, resulting in a high GF at low strain levels for the E-skin. Hence, we demonstrated the great potential of the sensor for applications in artificial intelligence, motion tracking, and health monitoring through experiments on fingers, wrists, and knees.

**Author Contributions:** R.W. designed and performed the research; S.F. and X.B. analyzed the data; Y.W. and Y.H. focused on the construction of models and the model analysis; R.W., S.F. and C.L. wrote the paper. M.H. and Y.Z. supervised the paper. All authors have read and agreed to the published version of the manuscript.

**Funding:** This work was supported by the National Natural Science Foundation of China (52173158, 32171725), Industrial prospect and key technology competition projects in Jiangsu Province (BE2021081), Postgraduate Research & Practice Innovation Program19 of Jiangsu Province (SJCX22_0056), Transformation Program of Scientific and Technological Achievements of Jiangsu Province (BA2019054, BA2021044).

**Institutional Review Board Statement:** Not applicable.

**Informed Consent Statement:** Not applicable.

**Data Availability Statement:** Data sharing not applicable.

**Conflicts of Interest:** The authors declare no conflict of interest.

**Abbreviations**

| | |
|---|---|
| E-skin | Electronic skin |
| PU | Polyurethane |
| AgNWs | Sliver nanowires |
| GF | Gauge factor |
| HDI | Hexamethylene diisocyanate |
| PTMEG | Polytetramethylene ether glycol |
| TBBPA | 3,5,3′,5′-Tetrabromobisphenol A |
| PG | Propyl gallate |
| DBTDL | Dibutyltin dilaurate |

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
