# Peer review of "A Transparent, and Self-Healable Strain-Sensor E-Skin Based on Polyurethane Membrane with Silver Nanowires"

_coatings, doi:10.3390/coatings13050829_

Round 1
Reviewer 2 Report
The article fits the journal and it is written quite well.
Some corrections:
a) Add a statistical analysis section
b) Lack of important references, such as:
Prabhakar, P.K., Raj, S., Anuradha, P.R., Sawant, S.N. and Doble, M., 2011. Biocompatibility studies on polyaniline and polyaniline–silver nanoparticle coated polyurethane composite. Colloids and Surfaces B: Biointerfaces, 86(1), pp.146-153.
Xu, D., Su, Y., Zhao, L., Meng, F., Liu, C., Guan, Y., Zhang, J. and Luo, J., 2017. Antibacterial and antifouling properties of a polyurethane surface modified with perfluoroalkyl and silver nanoparticles. Journal of Biomedical Materials Research Part A, 105(2), pp.531-538.
Baldelli, A., Esmeryan, K.D. and Popovicheva, O., 2021. Turning a negative into a positive: Trends, guidelines and challenges of developing multifunctional non-wettable coatings based on industrial soot wastes. Fuel, 301, p.121068.
Baldelli, A., Ou, J., Barona, D., Li, W. and Amirfazli, A., 2021. Sprayable, superhydrophobic, electrically, and thermally conductive coating. Advanced Materials Interfaces, 8(2), p.1902110.
and more.
Quite good, minor revisions.
Reviewer 3 Report
In this work, in order to solve the current problems, elastic and highly transparent polyurethane (PU) was first synthesized as the strain substrate following by the coating of sliver nanowires (AgNWs), which provided the strain-electric signal conversion path. The results of this manuscript are interesting and it can be accepted after following revisions.
1- English of the manuscript needs polishing.
2- The depth of technical discussions needs to go much further.
3- The literature review is required to be enriched by citing and commenting on the below articles focusing on strain sensing performance of nanocomposites: [Composites: Part A 163 (2022) 107244]; [Acta Materialia 230 (2022) 117870].
4- The percentage of nanopartcile in each experiment should be clearly stated.
5- Is it possible to study the effect of nanoparticle percentage on the stress-strain curves of recycled AgNWs/PU?
6- A brief discussion about important parameters affecting the stress-strain curves of AgNWs/PU should be added to the introduction. One of the main factors affecting the nanocomposite behavior is agglomeration of nanoparticles as mentioned in the following reference: [International Journal of Engineering Science 157 (2020) 103392].
7- What are the limitations of your experiments and research data?
English of the manuscript needs polishing.
Round 2
Reviewer 1 Report
I agree with the revisions for the manuscript done by the authors. No more comments, the author has incorporated good changes in the manuscript.